# Learning from discriminative feature feedback

**Sanjoy Dasgupta, Akansha Dey, Nicholas Roberts**
Department of Computer Science and Engineering
University of California, San Diego
dasgupta@eng.ucsd.edu,n3robert@ucsd.edu,a1dey@ucsd.edu

**Sivan Sabato**
Department of Computer Science
Ben-Gurion University of the Negev
sabatos@cs.bgu.ac.il

## Abstract

We consider the problem of learning a multi-class classifier from labels as well as simple explanations that we call *discriminative features*. We show that such explanations can be provided whenever the target concept is a decision tree, or can be expressed as a particular type of multi-class DNF formula. We present an efficient online algorithm for learning from such feedback and we give tight bounds on the number of mistakes made during the learning process. These bounds depend only on the representation size of the target concept and not on the overall number of available features, which could be infinite. We also demonstrate the learning procedure experimentally.

## 1   Introduction

Communication between humans and machine learning systems has typically been restricted to labels alone. A human provides labels for a data set and in return gets a classifier that predicts labels of new instances. This is a lot more rigid than the way humans learn. The field of *interactive learning* explores richer learning frameworks in which the machine engages with the human (or other sources of annotation) while learning is taking place, and the communication between the two is allowed to be more varied. A key question in this enterprise is whether such interaction can overcome fundamental algorithmic and statistical barriers to learning.

The one interactive framework that has perhaps been explored the most thoroughly is *active learning of classifiers*. Here, the learning machine begins with just a pool of unlabeled data, and interacts with a human by asking for labels of specific points. By adaptively focusing on maximally informative points, it can sometimes dramatically reduce the amount of labeling effort. Indeed, over the past two decades, a number of active learning algorithms have been designed that provably require only logarithmically as many labels as random querying, or otherwise reduce the labeling requirement considerably, in a variety of canonical settings, e.g. [1, 7].

In this paper, our interest is in feedback that goes beyond labels, to *simple explanations* of a particular type. Imagine, for instance, that you have just finished watching a movie in your living room, and your electronic assistant—Siri, Alexa, or one of their colleagues—asks "Did you like the movie?" You dutifully reply "yes", in effect providing a labeled data point. But then you casually add "I really like John Hurt". This last piece of information is spontaneous and does not require extra effort. But it can be far more useful, for the purposes of determining what movie to recommend next, than a mere thumbs-up or thumbs-down label. This kind of explanation could be called *feature feedback*, because it helps identify relevant features in a high-dimensional space of rationales for user preferences.

Feedback of this form has been used effectively in information retrieval [4, 13, 6] and in computer vision [10, 9, 8]. In general, however, these systems have been geared towards specific applications, and it is of interest to study such feedback rigorously, in a more abstract setting.

## 1.1 Predictive versus discriminative feature feedback

There has been a significant amount of work on what might be called *predictive feature feedback*. Suppose, for example, that in a document classification task, a labeler assigns each document $x$ to a category $y$ ("sports", "politics", and so on). While making this determination, the labeler might also be able to highlight a few words that are highly indicative of the label (e.g. "Congress", "filibuster"). This kind of auxiliary feedback has been explored in a variety of empirical studies, for text and image, with promising results [4, 13, 6, 14, 8]. More recently, the theoretical results of [12] have shown that such feedback can improve the rate of convergence of learning.

In this paper, we study an alternative setting that we call *discriminative feature feedback*. Consider a computer vision system that is learning to recognize different animals. Whenever it makes a mistake—classifies a "zebra" as a "horse", say—a human labeler ("the teacher") corrects it. While doing this, the labeler can also, at little extra cost, highlight a part of the image (the stripes, for instance) that distinguishes the two animals. Work on recognizing different species of birds, for instance, has used this sort of feedback effectively [3].

This kind of discriminative feedback is quite different from the predictive feature feedback of earlier work. In the document example, the feedback yields *predictive features*: the presence of the word "filibuster" is a moderately-strong predictor of the label "politics". In contrast, discriminative features are not necessarily predictive for the entire class. In the animal example above, "stripes" are not predictive of the class of zebras, since many animals have stripes. But they do distinguish zebras from horses. Thus discriminative feedback can be advantageous in a multi-class setting: the feature need only differentiate between two classes, rather than separating one entire class from all the others. In our abstract model, we relax this even further by requiring only that a discriminative feature separate one *subcategory* of a certain label from a subcategory of a different label.

## 1.2 Contributions

Our first contribution is to formalize one particular type of discriminative feature feedback. Human explanations, even simple ones, are rife with ambiguity and thus it is important to design explanatory mechanisms that have precise semantics.

Next, we present a simple and efficient learning algorithm that uses this feedback for multi-class classification. It operates in the online learning framework: at each point in time, a new data point arrives, it is classified by the learner's current model, and then feedback is received if this classification is incorrect. We show that the algorithm provably converges to the correct concept, and we provide a tight mistake bound on the total number of errors it makes over the lifetime of the learning process.

What concept classes are learned by our algorithm? We show that it can efficiently learn any multi-class concept that can be expressed as a decision tree, with a mistake bound that is quadratic in the number of leaves of the tree. More generally, it works when the target concept is expressible using a particular multi-class version of DNF (disjunctive normal form, OR-of-ANDs) formulas that we call *separable DNF*. In the setting of binary classification, disjunctive normal form concepts have proved to be computationally intractable to learn under standard supervised learning [11, 5], which is a bit troubling since humans seem to use such concepts quite naturally. But the hardness results apply to situations where the only feedback on any example is its label. With richer feedback, learning becomes easy and efficient.

The model learned by the algorithm is a logical combination of features obtained during feedback, a sort of decision list where each individual entry in the list is a conjunction. The mistake bound has no dependence on the overall number of available features, which could potentially be infinite. As a result this methodology can be used to build classifiers based on vast pools of low-level named features, which neural nets are increasingly able to provide.

Lastly, we demonstrate the learning procedure experimentally.

### 1.3 Other related work

Related to our work is the *infinite attribute model* of Blum [2], which introduces an online learning framework in which the goal is to learn a standard classifier, such as a linear separator, in situations where the total number of available features is infinite. Earlier mistake bounds for online learning, such as those for the Perceptron and Winnow algorithms, had some dependence on the number of features. It was shown in [2] that this dependence can be removed if, for any given example $x$, only finitely many features are present. These can, for instance, be thought of as the features that are perceptually most salient. In our paper, we again consider an online learning scenario and give a mistake bound that has no dependence on the overall number of available features. In our case, this arises naturally as a by-product of constructing classifiers from feature feedback, which is a different mechanism technically but does ultimately connect back to perceptual salience.

A different type of feedback was studied in [15], for the specific purpose of learning DNF formulas. The learner is allowed to make queries in which it provides two data points from the same class and asks how many terms in the DNF formula they both satisfy.

## 2 The feedback model

We introduce our formal model by means of an example. Let's say the goal is to learn a classifier that takes as an input example a description of an animal—given by a set of features describing where it lives, what it eats, its appearance, and so on—and then classifies it as `mammal`, `bird`, `reptile`, `amphibian`, or `fish`.

The learning takes place in rounds of interaction. On each round,

- A new animal is presented, e.g.,

    *seahorse*

- The learner classifies this instance using its current model.

    - E.g., it (mistakenly) says:

        ≫ `mammal`

    - In addition, the learner provides an example of an animal it has seen previously, that it considers to be similar to the new animal, and that belongs to the predicted class. E.g.,

        ≫ *seahorse* is similar to *horse*, which is a `mammal`.

- The labeler responds if the prediction is incorrect.

    - The labeler provides the correct class.

        > Correct class: `fish`

    - In addition, the labeler provides a distinguishing feature between the instance being classified and the instance that the learner suggested as a similar example.

        > Distinguishing feature between *seahorse* and *horse*: **lives-in-water**

Here the feature, **lives-in-water**, doesn't distinguish all fish from all mammals—some mammals do live in water, for instance—but does distinguish a group of fish that includes seahorses from a group of mammals that include horses.

We now formalize the semantics of this kind of interaction.

### 2.1 An abstract model of interaction

Let $c^*$ be the target concept to be learned, where $c^*$ is a mapping from the input space $\mathcal{X}$ (e.g., animals) to a finite label space $\mathcal{Y}$ (e.g., {`mammal`, `bird`, `reptile`, `amphibian`, `fish`}). The learner has access to a set of Boolean features $\Phi$ on $\mathcal{X}$, and expresses concepts in terms of these. For instance, in the example above, one of the features $\phi \in \Phi$ is **lives-in-water**.

Informally, the concepts we can handle are those in which: the data has some unknown underlying clusters (e.g., "regular land mammals", "egg-laying mammals", "marine mammals", "bony fish", "cartilaginous fish", etc); each label corresponds to a union of some of these clusters; and any two clusters with different labels can be separated by a single feature.

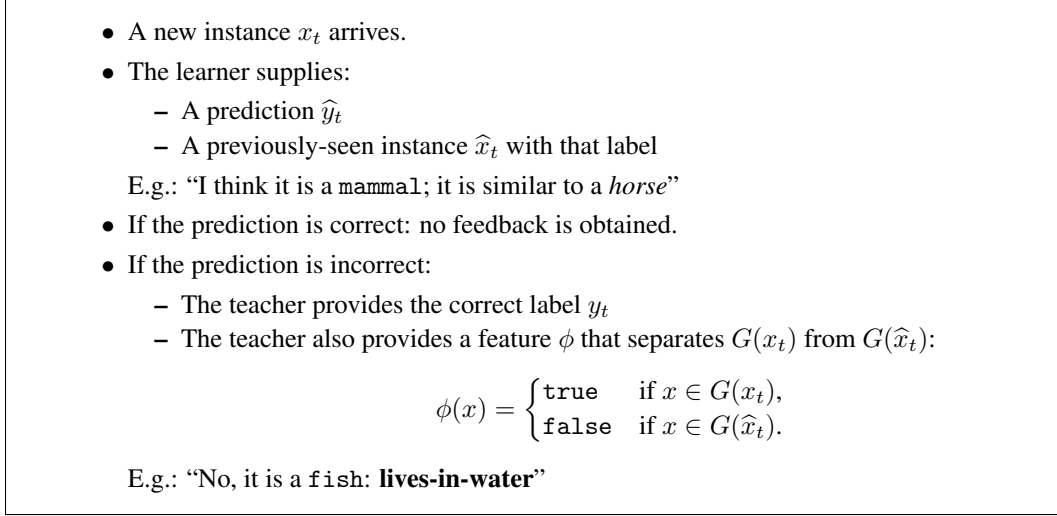

- A new instance $x_t$ arrives.
- The learner supplies:
  - A prediction $\widehat{y}_t$
  - A previously-seen instance $\widehat{x}_t$ with that label

  E.g.: "I think it is a `mammal`; it is similar to a *horse*"
- If the prediction is correct: no feedback is obtained.
- If the prediction is incorrect:
  - The teacher provides the correct label $y_t$
  - The teacher also provides a feature $\phi$ that separates $G(x_t)$ from $G(\widehat{x}_t)$:

  $$\phi(x) = \begin{cases} \texttt{true} & \text{if } x \in G(x_t), \\ \texttt{false} & \text{if } x \in G(\widehat{x}_t). \end{cases}$$

  E.g.: "No, it is a `fish`: **lives-in-water**"

Figure 1: Round $t$ of interaction.

Formally, we assume that $\mathcal{X}$ can be represented as the union of $m$ sets, $\mathcal{X} = G_1 \cup G_2 \cup \cdots \cup G_m$, possibly overlapping. This representation (that is, the identity of the sets in the union) is unknown to the learner, and satisfies the following:

- Each of the sets is pure in its label: for each $i$, there exists a label $\ell(G_i) \in \mathcal{Y}$ such that

  $$c^*(x) = \ell(G_i) \quad \text{for all } x \in G_i.$$

  (It follows that two sets $G_i, G_j$ can intersect only if they have the same label.)
- Any two sets $G_i, G_j$ with $\ell(G_i) \neq \ell(G_j)$ have a *discriminating feature*: there is some $\phi \in \Phi$ and $b \in \{0, 1\}$ such that

  $$\phi(x) = \begin{cases} b & \text{if } x \in G_i, \\ \neg b & \text{if } x \in G_j. \end{cases}$$

  For instance, if $G_i$ is the cluster of land mammals and $G_j$ is the cluster of cartilaginous fish, then one possible separating feature is **lives-in-water**.

As we discuss below, a representation that satisfies the assumptions above naturally exists if the set of Boolean features $\Phi$ is sufficiently rich. We place no restrictions on the number of features, which can even be infinite. Moreover, since the algorithm and the mistake bound have no dependence on the number of features in $\Phi$, the requirement that a *single* feature be used to separate sets is not restrictive: one can always include in $\Phi$ "single features" that are combinations of other basic features.

## 2.2 The learning protocol

Figure 1 shows how the $t$th round of interaction proceeds. In the protocol, $G(x)$ is the set (one of $G_1, \ldots, G_m$) containing $x$. If there are multiple such sets, it is some particular choice. Thus $G : \mathcal{X} \to \{G_1, \ldots, G_m\}$ and $x \in G(x)$.

To reiterate, the labeler does not need to provide features that separate entire classes from one another. Rather, these features just need to separate *a subgroup of one class*, containing $x_t$, from a subgroup of another class, containing $\widehat{x}_t$. These subgroups might just be the singletons $x_t$ and $\widehat{x}_t$, in which case the labeler need only distinguish these two specific instances. But it is reasonable to expect that a labeler will attempt to find features that are fairly general, in which case these subgroups will be somewhat larger. The clusters in the feedback formalism reflect the *level of categorization* at which the labeler is operating. They are allowed to be of arbitrary size; however, the complexity of the algorithm we later present will depend upon the total *number* of clusters. If there are $m$ clusters, the total number of mistakes made by the algorithm will be bounded by $m^2$.

## 2.3 When do the assumptions hold?

Our assumptions posit that there is some representation of the domain as purely-labeled sets that can be separated by single features, and that this representation is the one used implicitly by the teacher when providing feedback to the learner. An important question is when these assumptions hold. Moreover, since the number of sets in the representation determines the mistake bound of our learner, we would like to identify when this number is small.

First, note that whenever the set of features $\Phi$ is sufficiently rich that any two instances $x, x'$ differ on at least one feature, we can always have a representation based on singleton sets which satisfies the assumptions. In this sense, the model is non-parametric, and allows *any* target concept. However, the complexity of learning depends upon the number of sets, which we are denoting by $m$, and would be trivial if $m = |\mathcal{X}|$. Therefore, we would like to identify concepts which admit $m \ll |\mathcal{X}|$.

One case in which this holds is when the target concept $c^*$ can be expressed as a decision tree with $m$ leaves, using features in $\Phi$. We can define the set of examples $x \in \mathcal{X}$ that reach the $j$-th leaf to be a cluster $G_j$, so that there are $m$ clusters. This way, any two clusters $G_i$ and $G_j$ can be separated by a single feature: the feature $\phi$ at the internal node that is the lowest common ancestor of leaves $i$ and $j$.

More generally, discriminative feedback with $m$ clusters is possible if and only if the target concept is expressible as a particular kind of multi-class disjunctive normal form formula over the features in $\Phi$, that we call *separable-DNF*, and that has at most $m$ terms.

**Definition 1 (Separable-DNF concept)** *A separable-DNF concept over features $\Phi$ is a concept $c^* : \mathcal{X} \to \mathcal{Y}$ such that each individual class $y \in \mathcal{Y}$ is characterized by a DNF formula $F_y$, where $x$ satisfies $F_y$ if and only if $c^*(x) = y$, and:*

- *The literals in each $F_y$ are individual features from $\Phi$ or negations of such features.*
- *For any $y \neq y'$, denote $F_y = F_{y,1} \vee F_{y,2} \vee \cdots$ and $F_{y'} = F_{y',1} \vee F_{y',2} \vee \cdots$, where the $F_{y,i}$ and $F_{y',i}$ are conjunctions. Then for any $F_{y,i}$ and $F_{y',j}$, there is some feature $\phi \in \Phi$ that is in both conjunctions, but with opposite polarity.*

*We say that the separable-DNF concept is of size $m$ if the total number of conjunctions $F_{y,i}$ is $m$.*

**Lemma 1** *Target concept $c^*$ can be represented using $G_1, \ldots, G_m$ which satisfy the assumptions in Section 2.1 if and only if $c^*$ is a separable-DNF concept of size $m$ over features $\Phi$.*

The proof is deferred to Appendix A.

## 2.4 DNF formulas for binary classification

There is a large body of work on learning disjunctive normal form formulas for *binary* classification. These concepts don't exactly fall into the framework above because they are asymmetric: there is a DNF formula for positive instances, and everything else is a negative instance. In Appendix B, we provide a simple variant of our learning algorithm specifically for this case. For a target DNF formula with $m$ terms, each containing $k$ literals, it makes at most $m(k + 1)$ mistakes.

## 3 A mistake-bounded learning algorithm

We now show that under the setting of discriminative feature feedback, there is an efficient learning algorithm that reaches the target concept $c^*$ after making at most $m^2$ mistakes.

### 3.1 The algorithm

The algorithm is shown in Figure 2. It maintains:

- a list $L$ of some of the instances seen so far;
- for each item $x$ in this list, its label as well as a conjunction $C[x]$ that holds true for all of the cluster $G(x)$; and
- a default instance and label $(x_o, y_o)$ to apply to examples that violate all conjunctions of $L$.

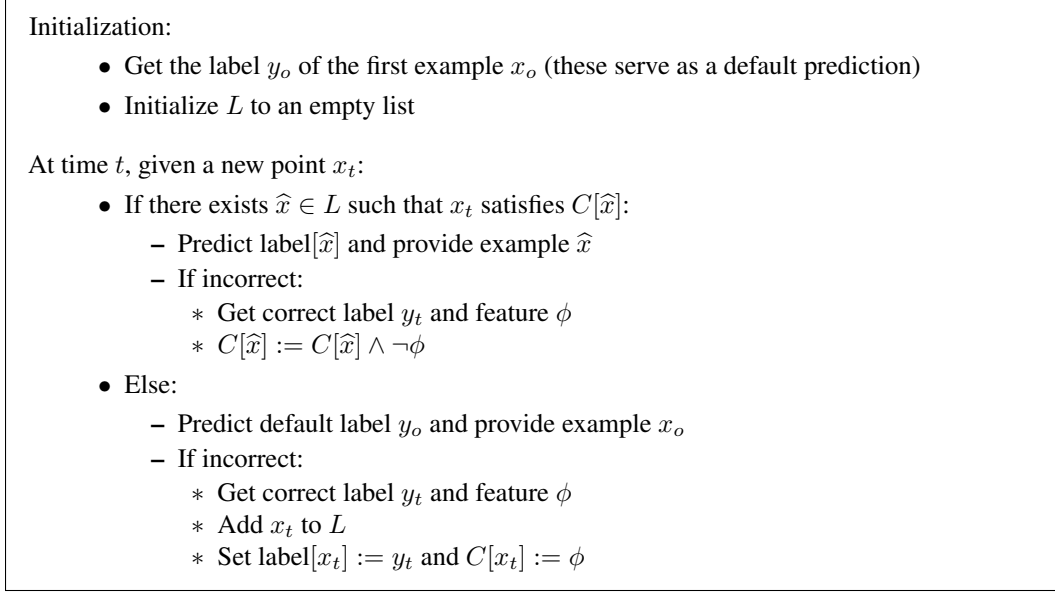

Initialization:

- Get the label $y_o$ of the first example $x_o$ (these serve as a default prediction)
- Initialize $L$ to an empty list

At time $t$, given a new point $x_t$:

- If there exists $\widehat{x} \in L$ such that $x_t$ satisfies $C[\widehat{x}]$:
  - Predict label$[\widehat{x}]$ and provide example $\widehat{x}$
  - If incorrect:
    * Get correct label $y_t$ and feature $\phi$
    * $C[\widehat{x}] := C[\widehat{x}] \wedge \neg\phi$
- Else:
  - Predict default label $y_o$ and provide example $x_o$
  - If incorrect:
    * Get correct label $y_t$ and feature $\phi$
    * Add $x_t$ to $L$
    * Set label$[x_t] := y_t$ and $C[x_t] := \phi$

Figure 2: An algorithm that learns from discriminative feature feedback.

The conjunctions $C[x]$ are built entirely out of features obtained from the teacher. We denote by $\mathcal{C}_L = \{C[x] : x \in L\}$ the set of conjunctions for the examples in $L$.

## 3.2 Mistake bounds

**Theorem 2** *Suppose that the algorithm of Figure 2 is used to learn a target concept $c^*$, and the teacher provides discriminative feature feedback corresponding to a representation $G_1, \ldots, G_m$ which satisfies the assumptions in Section 2.1. Then the total number of mistakes made over the lifetime of the algorithm is at most $m^2$.*

We prove this in several steps. First, we note two key invariants of the algorithm:

(I-1) Any item $x \in L$ has been seen in a previous round.

(I-2) For any $x \in L$, label$[x]$ is correct and every point in $G(x)$ satisfies $C[x]$.

The first invariant is trivial. The second invariant holds since the label is provided by the teacher, and the conjunction $C[x]$ is a conjunction of literals taken from the teacher. These literals are all satisfied by $G(x)$, as defined in the learning protocol in Figure 1.

Next, we show that $L$ contains at most one representative per group $G_i$.

**Lemma 3** *For any distinct $x, x' \in L$, we have $G(x) \neq G(x')$.*

PROOF: The only time a new $x$ is added to $L$ is in situations when it doesn't satisfy any of the conjunctions in $L$, and is therefore not in any of the corresponding $G(\cdot)$, as per invariant (I-2). $\square$

The above observations allow deriving a connection between the size of the conjunctions in $\mathcal{C}_L$ and the number of mistakes the algorithm makes.

**Lemma 4** *Let $B$ be an upper bound on the total number of literals in any conjunction in the list. Then the number of mistakes made by the algorithm is at most $mB$.*

PROOF: On each mistake of the algorithm, either: (i) an existing $x \in L$ has its conjunction restricted by an additional literal or (ii) a new item $x$ is added to $L$ with a conjunction of size 1. Thus, the total number of literals in conjunctions in $L$ is equal to the number of mistakes made by the algorithm. By assumption, each conjunction in $L$ has $\leq B$ literals, and by Lemma 3 there are at most $m$ such conjunctions. $\square$

We can now prove the mistake bound stated above.

**Proof of Theorem 2:**   We show that each conjunction in $L$ has at most $m$ literals as follows. First, any conjunction $C[\widehat{x}]$ always starts with a single literal. Subsequently, the conjunction is extended only when some instance $x_t$ appears that satisfies the conjunction and yet has $G(x_t) \neq G(\widehat{x})$. In this case, one literal is added to $C[\widehat{x}]$ and thereafter no instance in $G(x_t)$ satisfies the conjunction. Since there are $m$ different sets $G_i$, it follows that there can be at most $m-1$ rounds in which $C[\widehat{x}]$ is extended. Thus $C[\widehat{x}]$ has at most $m$ literals. The mistake bound now follows from Lemma 4. $\square$

Further, we show that this mistake bound is nearly tight, as stated in the following theorem and proved in Appendix C.

**Theorem 5** *The worst-case mistake bound of the algorithm in Figure 2, assuming discriminative feature feedback with $m$ clusters, is at least $(m-1)(m-2)/2$.*

It is also possible to derive a mistake bound which depends on the total number of features that the teacher uses during the interaction. This bound is useful if the teacher uses a small number of different features to discriminate between clusters. This can happen if the teacher attempts to reuse features, or if the target conjunction is sparse.

**Theorem 6** *Under the same assumptions as in Theorem 2, if the teacher uses at most $k$ features during the running of the algorithm, then the total number of mistakes made over the lifetime of the algorithm is at most $km$.*

PROOF: If the teacher uses at most $k$ different features during the run of the algorithm, then each conjunction in $L$ uses at most $k$ literals: Due to invariant (I-2), each conjunction is satisfied by at least one instance, thus it cannot include both a feature and its negation. The mistake bound follows from Lemma 4. $\square$

# 4   An illustrative experiment

The ZOO data set from the UCI ML repository contains information on 101 animals: for each, one of seven labels (`mammal`, `bird`, `reptile`, `fish`, `amphibian`, `insect`, `other`) as well as 21 Boolean features. The goal is to learn a classifier that predicts the label from the 21 features.

The learning algorithm of Figure 2 can potentially return different classifiers depending on the particular ordering of the data points. Figure 3 shows the result of one run. The starting example, and thus default prediction, is *frog* (class `amphibian`), and a total of 14 mistakes are made before convergence. On each mistake, a separating feature is chosen *at random* from those that distinguish the two instances ($x_t$ and $\widehat{x}$ in the notation of Figure 2). A human teacher would perhaps make more judicious feature choices.

The first table in the figure shows the prediction and similar example provided by the learner (columns 2 and 3) in each mistake round, as well as the correct label and discriminative feature provided by the labeler (columns 4 and 5). The final decision list $L$ is shown below. Recall that an instance is classified by first going through all the conjunctions in $L$ and then falling through to the default prediction if there is no match.

Notice that the list $L$ contains exactly 14 literals in all, one from each mistake. Also, we know from Lemma 3 that each of the "underlying groups" contributes at most one rule in $L$. Therefore, the method of choosing features implicitly divides the class `reptile` into at least two groups, which appear to correspond to land reptiles and aquatic reptiles, and divides the somewhat nebulous class `other` into two groups, exemplified by *worm* and *lobster*.

# 5   Conclusions and further directions

There is potential for enhancing the scope, robustness, and ease-of-use of learning systems by having them learn from simple explanations, and in turn explain their predictions. A crucial part of this enterprise is to identify and formalize simple explanatory mechanisms, and to study how they can be

| Instance | Prediction | Similar instance | True label | Discriminating feature |
|---|---|---|---|---|
| frog | | | amphibian | (default prediction) |
| worm | amphibian | frog | other | not(backbone) |
| girl | amphibian | frog | mammal | milk |
| herring | amphibian | frog | fish | zero-legs |
| seasnake | fish | herring | reptile | not(fins) |
| hawk | amphibian | frog | bird | two-legs |
| tortoise | amphibian | frog | reptile | not(aquatic) |
| termite | other | worm | insect | six-legs |
| lobster | amphibian | frog | other | not(backbone) |
| ladybird | reptile | tortoise | insect | six-legs |
| honeybee | other | lobster | insect | airborne |
| housefly | amphibian | frog | insect | not(aquatic) |
| flea | other | lobster | insect | breathes |
| seasnake | amphibian | frog | reptile | tail |
| newt | reptile | seasnake | amphibian | not(zero-legs) |

---

Final decision list ($L$):

- not(backbone) AND not(six-legs) $\implies$ other (*worm*)
- milk $\implies$ mammal (*girl*)
- zero-legs AND fins $\implies$ fish (*herring*)
- two-legs $\implies$ bird (*hawk*)
- not(aquatic) AND not(six-legs) $\implies$ reptile (*tortoise*)
- not(backbone) AND not(airborne) AND not(breathes) $\implies$ other (*lobster*)
- not(aquatic) $\implies$ insect (*housefly*)
- tail AND zero-legs $\implies$ reptile (*sea snake*)
- ELSE: amphibian (*frog*)

Figure 3: The 14 mistakes made on the ZOO data and the final classifier.

handled algorithmically. This paper introduces a novel type of explanation that is fairly intuitive, and shows that it lends itself to simple learning algorithms with rigorous performance guarantees. It is a first step, and a wide range of open problems remain.

For this particular feedback scheme, what happens if the labeler is noisy or careless? We know from Section 2 that the scheme can in principle handle any concept, and thus noisy feedback is unlikely to derail convergence: if necessary, for instance, a noisily-labeled point can be in a cluster of its own. A more subtle issue is how much the mistake bound can blow up as a result of noise.

The decision lists produced by the algorithm of Figure 2 are accompanied by reassuring guarantees. However, is it possible to retain those guarantees while producing a model that is more concise? This open problem is particularly important in cases where the lists are intended to be interpretable.

## Acknowledgements

This research was supported by National Science Foundation grant CCF-1813160, and by a United-States-Israel Binational Science Foundation (BSF) grant no. 2017641. Part of the work was done while SD and SS were at the "Foundations of Machine Learning" program at the Simons Institute for the Theory of Computing, Berkeley.

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
