[Reviews · NeurIPS 2018]

Reviewer 1



The paper provides an interesting view on interactive machine learning, in which the machine elicits explanations to the user and incrementally builds a logical model incorporating these explanations. It proves that under certain rather reasonable conditions on the behaviour of the user, the error of the system can be bounded in terms of some intrinsic clustering of the data. The experimental evaluation is only for illustrative purposes. My main concern with this work is that it seems to shift the entire effort of producing a discriminative model to the user. The system basically incrementally builds the model incorporating rules generated by the user. I would expect a learning system to provide some sort of proactiveness in building the learning model, and rely on the user only occasionally (the proposed system queries the user for each prediction), possibly asking for explanations as proposed in this paper. AFTER REBUTTAL: I still believe the role of the learning algorithm is too passive here, but I agree with the authors that the type of feedback required is not too much harder than labeling, and that in the restricted setting they are addressing they do have a message.

Reviewer 2



The paper describes an online active learning protocol (and a classification algorithm) that involves additional discriminative feature feedback. In other words, unlike the classical online protocol, upon prediction of the class label the learner receives not only the correct label, but also some discriminative feature of the class whenever it errs. In considered learning setting, the feature space is binary and a class is represented by the union of some unknown clusters, where each cluster in its turn represents some specialization of the class (i.e. an attribute) and every two clusters can be discriminated by only one binary feature. The algorithm roughly operates by keeping around instances seen so far and constructing DNFs that representing clusters. The main contribution of the paper is the mistake bound for an algorithm that learns a subclass of DNFs in the described protocol. This mistake bound in the worst case scales quadratically in the total number of clusters. This seemingly comes in contrast with known results that learning DNFs is intractable, however, given only the label as a feedback. This paper demonstrates that learning is possible, however, in presence of additional feedback. The fact that the number of mistakes is bounded by the square of the number of clusters appears to match known bounds for multiclass online prediction (square in the number of classes).

Reviewer 3



Main Idea This paper analyzes a novel setting of interactive learning where the user gives a discriminative feature as feedback in addition to the labels. The setting is quite simple: we suppose data points form m clusters, where each cluster is pure in its label. The discriminative feature feedback is defined as the feature that differentiate two clusters. The paper shows that under this setting all possible classifiers can be represented as a certain kind of DNF, and gives an algorithm with regret at most m^2. Comments Overall, the paper is written nicely and easy to follow. However I am a bit concerned about its contribution in terms of theoretical results: The setting here seems quite simplified and resulting algorithm is not practical for most interested applications. E.g., in the Siri example in section 1, the feedback of human for why he likes the movie might be of many reasons, and the resulting feature space will be very large and sparse (and therefore there will be a large number of clusters). In this case it would be prohibitive to make up to m^2 mistakes. It is therefore important to analyze cases where (1) clusters are not pure, (2) human are not perfect, and (3) features are continuous. Minor comments 1. It will be interesting to know some lower bounds of a general algorithm in this setting. 2. Line 179: It should be "total number of conjunction terms [in] F_{y,i} is m" Quality The paper is well written and the analysis seems complete. Clarity The paper is mostly clear. Originality The paper is original. Significance The paper analyzes the important problem of learning with discriminative features. However, the setting seems too restricted and simplified. --Update-- Although the authors argue that their algorithm can work under even an infinite number of features, it is not shown that how the number of groups behaves in such settings. The paper can be more influential if, e.g., the authors can related the number of groups to sparsity conditions and/or logarithm of # features. I'm therefore not changing my review score. --update 2-- After seeing discussions I would raise my score by 1 in this situation.